# Endurance Runners with Intramyocellular Lipid Accumulation and High Insulin Sensitivity Have Enhanced Expression of Genes Related to Lipid Metabolism in Muscle

**DOI:** 10.3390/jcm9123951

**Published:** 2020-12-06

**Authors:** Saori Kakehi, Yoshifumi Tamura, Kageumi Takeno, Shin-ichi Ikeda, Yuji Ogura, Norio Saga, Takeshi Miyatsuka, Hisashi Naito, Ryuzo Kawamori, Hirotaka Watada

**Affiliations:** 1Department of Metabolism and Endocrinology Juntendo University Graduate School of Medicine, 2-1-1 Hongo, Bunkyo-ku, Tokyo 113-8421, Japan; skakei@juntendo.ac.jp (S.K.); t-kage@juntendo.ac.jp (K.T.); shin-ikeda@keio.jp (S.-i.I.); miyatsuka@juntendo.ac.jp (T.M.); kawamori@juntendo.ac.jp (R.K.); hwatada@juntendo.ac.jp (H.W.); 2Sportology Center, Juntendo University, Tokyo 113-8421, Japan; 3Institute of Health and Sports Science & Medicine, Juntendo University, Chiba 270-1695, Japan; yuji_ogura@marianna-u.ac.jp (Y.O.); n_saga@main.teikyo-u.ac.jp (N.S.); hnaitou@juntendo.ac.jp (H.N.); 4Department of Exercise Physiology, Graduate School of Health and Sports Science, Juntendo University, Chiba 270-1695, Japan; 5Center for Therapeutic Innovations in Diabetes, Juntendo University, Tokyo 113-8421, Japan; 6Center for Identification of Diabetic Therapeutic Targets, Graduate School of Medicine, Juntendo University, Tokyo 113-8421, Japan

**Keywords:** athlete’s paradox, intramyocellular lipid, insulin resistance, adiponectin receptor

## Abstract

Context: Endurance-trained athletes have high oxidative capacities, enhanced insulin sensitivities, and high intracellular lipid accumulation in muscle. These characteristics are likely due to altered gene expression levels in muscle. Design and setting: We compared intramyocellular lipid (IMCL), insulin sensitivity, and gene expression levels of the muscle in eight nonobese healthy men (control group) and seven male endurance athletes (athlete group). Their IMCL levels were measured by proton-magnetic resonance spectroscopy, and their insulin sensitivity was evaluated by glucose infusion rate (GIR) during a euglycemic–hyperinsulinemic clamp. Gene expression levels in the vastus lateralis were evaluated by quantitative RT-PCR (qRT-PCR) and microarray analysis. Results: IMCL levels in the tibialis anterior muscle were approximately 2.5 times higher in the athlete group compared to the control group, while the IMCL levels in the soleus muscle and GIR were comparable. In the microarray hierarchical clustering analysis, gene expression patterns were not clearly divided into control and athlete groups. In a gene set enrichment analysis with Gene Ontology gene sets, “RESPONSE TO LIPID” was significantly upregulated in the athlete group compared with the control group. Indeed, qRT-PCR analysis revealed that, compared to the control group, the athlete group had 2–3 times higher expressions of proliferator-activated receptor gamma coactivator-1 alpha (PGC1A), adiponectin receptors (AdipoRs), and fatty acid transporters including fatty acid transporter-1, plasma membrane-associated fatty acid binding protein, and lipoprotein lipase. Conclusions: Endurance runners with higher IMCL levels have higher expression levels of genes related to lipid metabolism such as PGC1A, AdipoRs, and fatty acid transporters in muscle.

## 1. Introduction

Skeletal muscle has been shown to adapt to various stimuli, including exercise. Indeed, endurance training enhances mitochondrial biogenesis, lipid and glucose transport, lipid oxidation, and oxidative capacity in muscle [1,2,3,4,5,6,7]. In addition, endurance-trained athletes who have high oxidative capacity and enhanced insulin sensitivity also have higher intramyocellular (IMCL) content [8,9]. These characteristics of the muscles in endurance-trained athletes are likely acquired as a long-term effect of training. It is supposed that gene expression levels are chronically altered by endurance training over the long term, which contributes to the altered characteristics in the muscles of endurance-trained athletes.

Several studies have compared global gene expression levels in skeletal muscle using microarrays before and after endurance training for 6 to 20 weeks [10,11,12,13]. However, only a few studies have included microarray analysis of muscles in athletes with long-term endurance training [14,15]. Stepto et al. [14] demonstrated that cyclists with long-term (> 5 years) endurance training have higher expressions of gene clusters related to mitochondrial/oxidative capacity than the control subjects. In addition, VO_2_ peak was correlated with clusters of mitochondrial, fat, and carbohydrate oxidation genes, respectively. Similarly, Wittwer et al. demonstrated that expression levels of genes related to glycolysis were also enhanced in a professional cyclist with 8 years of endurance training [15]. However, the overall gene profile in the skeletal muscle of endurance runners compared with control subjects has not been clarified yet.

Based on this background, we investigated the differences in gene expression profiles in skeletal muscle of endurance runners and control subjects using microarray analysis. In addition, we performed quantitative RT-PCR (qRT-PCR) to specifically measure gene expression levels related to IMCL accumulation and insulin sensitivity, which were clarified in our previous study [16].

## 2. Research Design and Methods

### 2.1. Subjects

Study subjects consisted of 7 endurance runners with average records and 8 healthy men with BMI < 23 kg/m^2^. The endurance runners’ personal records in 5000 m ranged between 14:33.70 and 15:50.00. They were in good health as determined by medical history, physical examination, and standard blood chemistry analyses. All subjects gave written informed consent to the study, which was approved by the Ethics Committee of Juntendo University.

### 2.2. Study Design and Measurement of Various Parameters

Regular exercise was prohibited from at least 3 days before the experiment day [16]. Each subject consumed a 3-day isocaloric normal-fat diet. Blood samples were taken after overnight fasting. A biopsy of the vastus lateralis muscle was also performed using a needle with local anesthesia. Muscle samples were immediately placed in the RNA-stabilizing reagent RNAlater (Qiagen, Hilden, Germany) and stored at −80 °C. Extraction of total RNA and DNA from muscle samples was performed as previously described [1,16]. Biochemical analyses of serum samples and total body fat content measurements were performed as described previously [1,16].

### 2.3. Proton Magnetic Resonance Spectroscopy

IMCL in the tibialis anterior muscle (TA) and soleus muscle (SOL) were measured after overnight fasting using ^1^H-magnetic resonance spectroscopy (VISART 1.5T EX V4.40; Toshiba, Tokyo), as described previously [1,16,17,18,19,20,21]. Voxels (1.2 × 1.2 × 1.2 cm^3^) were positioned in the TA or SOL avoiding visible interfascial fat and blood vessels. Imaging parameters were set as follows: repetition time of 1500 ms, echo time of 136 ms, acquisition numbers of 192, and 1024 data points over a 1000-kHz spectral width. After examination, the resonances were line fit using a mixed Lorentzian/Gaussian function. After corrections for T_1_ and T_2_ relaxations, the methylene signal intensity (S-fat), with peaks observed at approximately 1.25 parts/million (ppm), was quantified. IMCL was quantified by S-fat with the creatine signal at 3.0 ppm (Cre) as the reference and was calculated as a ratio relative to Cre (S-fat/Cre).

### 2.4. Hyperinsulinemic Euglycemic Clamp Study

A hyperinsulinemic euglycemic glucose clamp study with a target plasma glucose level of 95 mg/dL and an insulin infusion rate of 100 mU/m^2^/min was performed with an artificial pancreas (STG22; Nikkiso, Shizuoka, Japan), as reported previously [20]. The steady-state glucose infusion rate (GIR) was observed from 105 to 120 min after the beginning of the study and was used as a marker of peripheral insulin sensitivity [1,16,17,18,19,20,21].

### 2.5. DNA Microarray Analysis

Microarray analysis was performed with muscle samples from 7 control subjects and 5 endurance athletes, as described previously [1]. The transcript profile of each sample was determined using the Affymetrix Human U133 Plus 2.0 array with standard Affymetrix protocols (Affymetrix, Santa Clara, CA, USA). The data were summarized using Genespring GX software (Agilent Technologies, Santa Clara, CA, USA). The signal intensities for the beta actin (*ACTB*) and glyceraldehyde 3-phosphate dehydrogenase (*GAPDH*) genes were used as internal quality controls. To correct for variations between gene chips, the signal data (CEL files) were quantile normalized, with probe set intensities calculated using the MAS5 method. Genes identified as being absent across all samples using Genespring GX were excluded from further analyses. The gene expression profiles have been deposited in the Gene Expression Omnibus database under the accession number GSE155271.

### 2.6. DNA Microarray Data Analysis

Ward’s hierarchical two-way classification method was performed using differentially expressed genes and the subjects. A false discovery rate of less than 5% was considered significant. A cut-off value of 1.3 for fold change_ratios was used. Differentially expressed genes were also identified using gene set enrichment analysis (GSEA) with the Kolmogorov–Smirnov test performed on each sample group. Analysis was performed with Gene Ontology gene sets (c5.all.v2.5) and Reactome gene sets (c2).

### 2.7. qRT-PCR

Gene expression levels were examined using qRT-PCR (1). Primers were designed using Primer-BLAST.

### 2.8. Statistical Analysis

All data are expressed as means ± SD. Differences between groups were evaluated using the unpaired *t*-test. Statistical significance was set at *p* < 0.05.

## 3. Results

### 3.1. Characteristics of the Subjects

Table 1 shows the anthropometric data in the control and athlete groups. Total cholesterol was significantly higher in the athlete group than in the control group. Maximum oxygen uptake was numerically higher in the athlete group, but this difference was not statistically significant. As shown in Figure 1, IMCL levels in TA were significantly higher in the athlete group (*p* < 0.05). GIR and IMCL levels in SOL were comparable between the groups.

### 3.2. Hierarchical Clustering Analysis in the Control and Athlete Groups

We used DNA microarrays to profile the gene expression in skeletal muscle in 13 subjects, 5 from the athlete group and 8 from the control group. Hierarchical clustering analysis based on gene expression demonstrated that gene expression patterns were not clearly divided into control and athlete groups, suggesting that differences in gene expression patterns between controls and athletes identified in this study do not surpass differences in gene expression patterns across individuals (Figure 2).

### 3.3. Gene Expression Profiles of Muscle in the Control and Athlete Groups

We identified 613 genes or probes whose expressions were significantly upregulated or downregulated in athletes compared with controls (*p* < 0.05, fold change > |1.3|). We identified 286 downregulated genes and 330 upregulated genes in the athlete group compared to the control group (Figure 3A, Appendix A). We did not find genes with more than a 3-fold change. The upregulated gene with the highest fold change (2.38) was nicotinamide riboside kinase 2 (*NMRK2*). The downregulated gene with the lowest fold change (−2.68) was pyruvate dehydrogenase kinase 4 (*PDK4*).

### 3.4. Gene Set Enrichment Analysis (GSEA) of Skeletal Muscle Tissue in the Control and Athlete Groups

In order to gain insight into the biological effects of gene expression patterns, we performed a GSEA with Gene Ontology gene sets (c5.all.v2.5) (Figure 3B,C, Appendix A) and Reactome gene sets (c2) (Appendix A). This analysis found 70 gene sets significantly upregulated in the control group (*p* < 0.05) (Figure 3B, Appendix A) and 13 gene sets significantly upregulated in the athlete group (*p* < 0.05) (Figure 3C, Appendix A). Given that one of the important characteristics of endurance-trained athletes is higher IMCL content [8,9], “RESPONSE TO LIPID” (Biological Process) was among the gene sets significantly upregulated in the athlete group compared to the control group (*p* = 0.045) (Figure 3C, Appendix A).

### 3.5. Gene Expression Analysis Using qRT-PCR in the Control and Athlete Groups

Previously, we found higher expressions of fatty acid transporters and fatty acid oxidation-related genes in muscle from nonobese nonathletes with high IMCL accumulation and high insulin sensitivity [16]. Here, we found that the “RESPONSE TO LIPID” gene set was significantly upregulated in the athlete group. Thus, we evaluated the expression of genes related to lipid metabolism.

Gene expression levels in fatty acid transporter 1 (*FATP1*), plasma membrane-associated fatty acid binding protein (*FABPpm*), and lipoprotein lipase (*LPL*), which regulate fatty acid uptake in muscle, were approximately 2 times higher in the athlete group than in the control group (*p* < 0.05) (Figure 4A). Furthermore, gene expression of proliferator-activated receptor alpha (*PPARA*), a key transcription factor for fatty acid oxidation; proliferator-activated receptor gamma coactivator-1 alpha (*PGC1A*), a master regulator of mitochondrial biogenesis; hydroxyacyl-CoA dehydrogenase trifunctional multienzyme complex subunit beta (*HADHB*), a mitochondrial beta oxidation enzyme; and adiponectin receptor 1 (*ADIPOR1*) and *ADIPOR2*, which mediate fatty acid oxidation and glucose uptake by adiponectin [22,23] were also significantly higher in the athlete group (Figure 4B).

## 4. Discussion

In the present study, we compared the gene expression levels in skeletal muscle and characteristics such as IMCL accumulation and insulin sensitivity between endurance runners and control subjects. IMCL levels in TA were approximately 2.5 times higher in the athlete group compared with the control group, while IMCL levels in SOL and GIR were comparable between the groups. In the microarray hierarchical clustering analysis, gene expression patterns were not clearly divided into control and athlete groups. In the GSEA with Gene Ontology gene sets, “RESPONSE TO LIPID” was significantly upregulated in the athlete group compared to that in the control group. The qRT-PCR analysis revealed that the athlete group had expression levels of several genes related to fatty acid transportation, including *FATP1*, *FABPpm,* and *LPL*, that were approximately 2 times higher than in the control group. In addition, genes related to fatty acid oxidation such as *PPARA, PGC1A, HADHB, ADIPOR1,* and *ADIPOR2*, were also higher in the athlete group.

In the present study, IMCL levels in TA were approximately 2.5 times higher in endurance athletes than in healthy controls, while IMCL levels in SOL and GIR were comparable between the groups. However, this is not surprising since IMCL is not always raised in endurance athletes [24,25]. A previous study also suggested that IMCL levels in TA are approximately 2 times higher in endurance runners compared with sprinters, while IMCL levels in SOL was comparable between the groups [21]. SOL is a muscle that is rich in type I fibers and has higher IMCL levels, mitochondria, and capillary density [26,27,28,29]; these characteristics are acquired with endurance training [28]. In fact, IMCL in endurance athletes was raised due to higher proportions of type I fiber [30]. On the other hand, TA contains a lower proportion of type I fibers than SOL [31]. Thus, it is possible that endurance training might induce IMCL accumulation in TA, but to a lesser extent in SOL.

We analyzed the gene expression profiles in skeletal muscle of endurance runners using microarray analysis. A previous study investigating the skeletal muscle gene profile of endurance cyclists with microarray analysis showed enhanced expression of gene clusters related to mitochondrial and oxidative capacity [14]. In the present study, as shown in the hierarchical cluster analysis, gene expression was not clearly divided by controls versus endurance athletes, suggesting that endurance runners are not significantly characterized by gene expression patterns in muscle. It may also be possible that control subjects in the present study had relatively higher maximum oxygen uptake than those of a previous study [14], leading to no clear significant differences in gene expression levels between the groups in the microarray analysis. Nonetheless, in the GSEA, “RESPONSE TO LIPID” was upregulated in the athlete group compared to that in the control group, suggesting that this gene cluster might be important in characterizing endurance runners among healthy young people.

It has been suggested that the oxidative capacity of skeletal muscle may be an important modulator of the association between IMCL accumulation and insulin sensitivity [8]. Previous studies have suggested that fatty acid influx into myocytes increases oxidative capacity in muscle. For example, it has been reported that endurance runners have increased FABPpm and LPL [32,33] expression levels, and free fatty acid uptake in skeletal muscle is enhanced in endurance athletes [34]. In addition, overexpression of FABPpm, FATP-1, and LPL increases fatty acid oxidative capacity and mitochondrial content in muscle [35,36,37]. Moreover, a high-fat diet stimulates mitochondrial fatty acid oxidation and increases muscle expression levels of PGC1A, HADHB, and CPT-1 in rodents [38]. Taken together, higher expression levels of genes including *FATP1*, *FABPpm*, and *LPL* may facilitate IMCL accumulation and promote oxidative capacity in muscle and may characterize the athlete’s paradox. 

In the present study, expression levels of PPARA, PGC1A, AdipoRs, and several fatty acid transporters were higher in the athlete group compared with the control group. PPARA is a transcriptional factor activated by fatty acids, leading to the induction of genes involved in fatty acid import and β-oxidation [39,40,41,42]. In addition, in vitro and in vivo studies have demonstrated that PGC1A overexpression results in increased expression of CD36, FABPpm, FATPs, mitochondrial content, and oxidative capacity in muscle [43]. We previously showed that overexpression of FABPpm in C2C12 myotubes results in enhanced expression levels of PPARA, PGC1A, AdipoR1, and AdipoR2, while fatty acid uptake and oxidation were both increased in the myotube [16]. PPARα and PGC-1α activation are partly regulated by adiponectin via AdipoR2 and AdipoR1, respectively [3,22]. Exercise training increases muscle AdipoRs [2,3,4], PGC1A, and PPARA expression [5,6,7]. Thus, it has been suggested that PPARA, PGC1A, AdipoRs, and fatty acid transporters are upregulated by endurance exercise and fatty acid incorporation into myocytes, leading to both higher IMCL levels and higher oxidative capacity in endurance athletes.

One limitation of the present study is that we included only male subjects. It has been reported that fatty acid metabolism in skeletal muscle differs by sex and that females have higher levels of IMCL than males. In addition, the number of subjects in the microarray analysis was small, so further study is required to identify specific features of gene expression patterns in endurance athletes. There was no significant difference in maximum oxygen uptake between the groups. Thus, if we included more unfit subjects as controls, we might have different results in IMCL levels and gene expression profiles. Finally, gene expression levels were evaluated in vastus lateralis muscle, while IMCL levels were measured in TA and SOL. Thus, it is still unclear how IMCL accumulation is associated with muscle gene expression in endurance athletes.

In conclusion, endurance runners had higher IMCL levels in TA but similar IMCL levels in SOL and insulin sensitivity as the heathy controls. These characteristics of endurance runners were associated with elevated expressions of genes related to lipid metabolism, including lipid entry (*FATP1, FABPpm,* and *LPL*), lipid oxidation (*HADHB*), and their upstream regulators (*PPARA, PGC1A, AdipoRs*) in skeletal muscle.

## Figures and Tables

**Figure 1 jcm-09-03951-f001:**
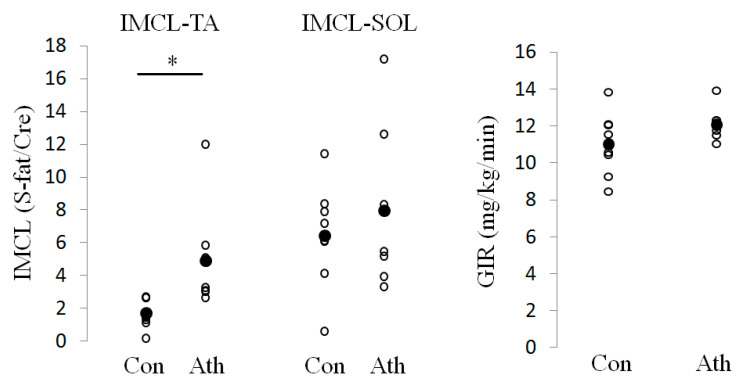
Absolute intramyocellular lipid (IMCL) and glucose infusion rate (GIR) levels in the control and athlete groups. Absolute values of GIR and IMCL in the control group (Con) and athlete group (Ath). The average in each group is shown as closed circle. * *p* < 0.05 Intramyocellular lipid, IMCL; tibialis anterior, TA; soleus, SOL; methylene signal intensity, S-fat; creatine signal, Cre.

**Figure 2 jcm-09-03951-f002:**
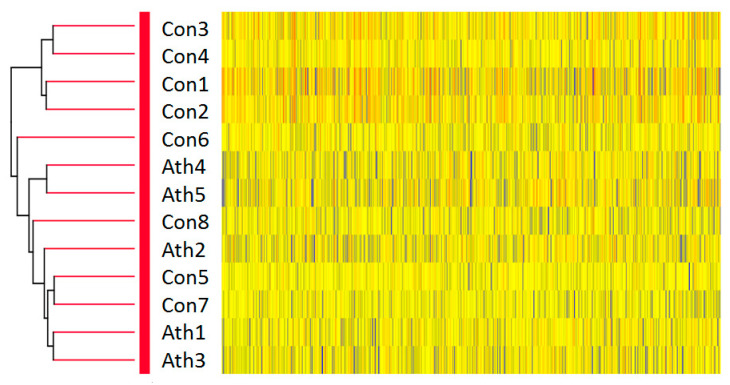
Hierarchical clustering of microarray data in the control and athlete groups. Euclidean distance was used as the metric. The tree was constructed using Ward’s method. Ath1–Ath5, Endurance athletes (*n* = 5); Con1–Con8 Control subjects (*n* = 8).

**Figure 3 jcm-09-03951-f003:**
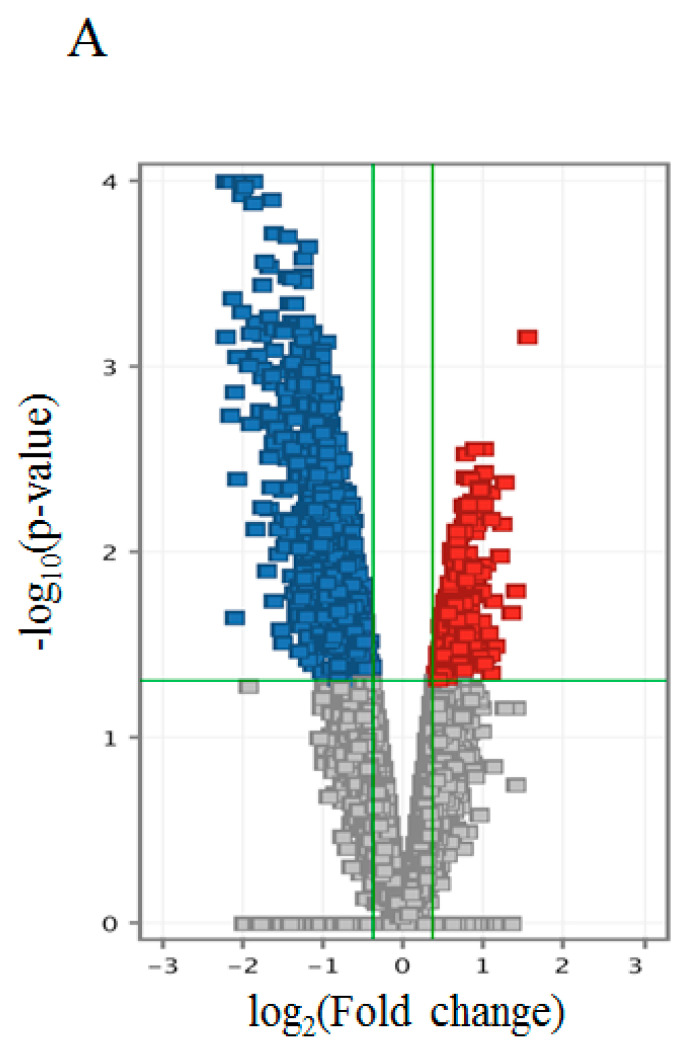
Microarray findings in the control and athlete groups. (**A**) Volcano plot comparing gene expression in the control versus athlete group. The x-axis indicates differential expression profiles plotted in fold-induction ratios on the log_2_ scale. The y-axis indicates the statistical significance of the difference in expression on the log_10_ scale. Genes upregulated in the athlete group are shown in red. Genes downregulated in the athlete group are shown in blue. (**B**) The top 20 significantly upregulated Gene Ontology gene sets (c5.all.v2.5) based on GSEA in the control group. (**C**) The top 20 significantly upregulated Gene Ontology gene sets (c5.all.v2.5) based on GSEA in the athlete group. Gene set enrichment analysis; GSEA, Normalized enrichment score; NES.

**Figure 4 jcm-09-03951-f004:**
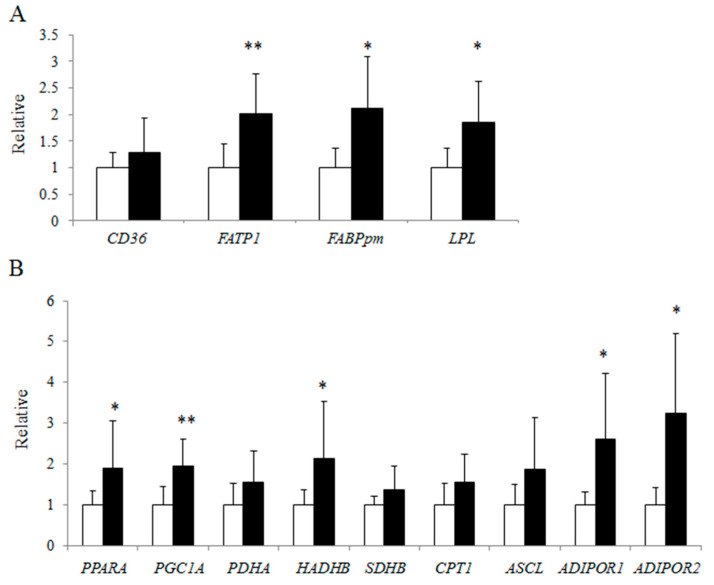
Gene expression analysis of skeletal muscle in the control and athlete groups using qRT-PCR. (**A**) Expression levels of fatty acid transporter in the control group (white bars) and the athlete group (black bars). Data are relative to the expression level in the control group, which was set to 1. (**B**) Expression levels of genes related to fatty acid β-oxidation in the control group (white bars) and the athlete group (black bars). Data are relative to the expression level in the control group, which was set to 1. * *p* < 0.05, ** *p* < 0.01. Values were obtained by normalization to a housekeeping gene (*ACTB*). Data are presented as means ± SD. *CD36*: CD36, *FATP1*: fatty acid transporter protein 1, *FABPpm*: plasma membrane-associated fatty acid binding protein, *LPL*: lipoprotein lipase, *PPARA*: peroxisome proliferator-activated receptor-α, *PGC1A*: proliferator-activated receptor gamma coactivator-1α, *PDHA*: pyruvate dehydrogenase-α; *HADHB*: hydroxyacyl-CoA dehydrogenase-β, *SDHB* succinate dehydrogenase subunit B, *CPT1*: carnitine palmitoyltransferase 1, *ACSL*: long chain acyl-CoA synthetase, *ADIPOR1*: adiponectin receptor 1, *ADIPOR2*: adiponectin receptor 2.

**Table 1 jcm-09-03951-t001:** Characteristics of the control and athlete groups.

	Control Group (*n* = 8)	Athlete Group (*n* = 7)
Age (years)	21.7 ± 1.4	21.6 ± 0.8
Body mass index (kg/m^2^)	19.7 ± 1.1	20.5 ± 1.1
Body fat (%)	12 ± 3.5	13.3 ± 3.0
Glucose (mg/dL)	83.4 ± 5.1	85.6 ± 7.1
Insulin (μU/mL)	2.8 ± 1.0	4.3 ± 1.2
Free fatty acids (mmol/L)	0.43 ± 0.16	0.49 ± 0.10
Total cholesterol (mg/dL)	169 ± 17.2	208.6 ± 33.3 *
High-density lipoprotein cholesterol (mg/dL)	59.4 ± 11.4	59.4 ± 10.9
Triglycerides (mg/dL)	62.2 ± 31.0	65.2 ± 20.3
Hemoglobin A1c (%)	4.7 ± 0.2	4.7 ± 0.2
High molecular weight adiponectin (µg/mL)	1.81 ± 1.15	1.89 ± 1.20
Maximum oxygen uptake (mL/kg/min)	52.2 ± 6.9	58.4 ± 3.9

Data are presented as means ± SD. * *p* < 0.05.

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
