# Peer review of "Endurance Runners with Intramyocellular Lipid Accumulation and High Insulin Sensitivity Have Enhanced Expression of Genes Related to Lipid Metabolism in Muscle"

_jcm, 2020, doi:10.3390/jcm9123951_

Round 1

Reviewer 1 Report

This manuscript by Kakehi, Tamura et al. measures intramyocellular lipid, VO2max, and insulin sensitivity in athletes and controls and tries to explain 'expected' differences in terms of measured alterations in gene expression levels in muscle. However, the research design appears funadamentally flawed as there is no statistical difference in VO2max nor insulin sensitivity between the 'athletes' and 'controls'.

With this lack of significance between groups what power calculations were performed?

There appear a lack of appreciation/mention that the athlete's paradox is not always evident ie. that IMCL is not always raised in athletes (eg. Bruce et al. 2003 JCEM; Bergman et al. 2018 Am J Physiol Endocrinol Metab) - and the results of this study are in agreement with this. An early study suggested IMCL was raised due to differences in fibre type (van Loon et al. 2004 Am J Physiol Endocrinol Metab), where type 1 fibres are known to have significantly higher IMCL. Also, given the differences between muscles, then the gene expression data should ideally be from the same muscle group.

Intramyocellular lipid methodology - there appears excessive references which repeat or do not contain full methods. It is still unclear - was this done on a 1.5T scanner? Given that a long TE of 136ms and short TR of 1500ms were used, then these measures of IMCL and Cr would be T1 and T2-weighted. Given that exercise training likely results in muscle microstructural differences that could result in different relaxation times in athletes and controls, then were T1 and T2 measured and corrected for? Also, how was 'signal intensity' measured - details of fitting procedure should be documented.

Minor points:

Fig 4 legend - should 'HR' be athletes?

Fig 1 - would be better to have as dot plots to be able to view the distribution.

Line 59 - it is normal to refer to surnames, not forenames.

Author Response

Responses to the comment of Reviewer 1

First of all, we would like to express our sincere thanks to you for identifying areas that needed corrections or modification. We would like to respond to your comment as described below.

This manuscript by Kakehi, Tamura et al. measures intramyocellular lipid, VO2max, and insulin sensitivity in athletes and controls and tries to explain 'expected' differences in terms of measured alterations in gene expression levels in muscle. However, the research design appears funadamentally flawed as there is no statistical difference in VO2max nor insulin sensitivity between the 'athletes' and 'controls'.

With this lack of significance between groups what power calculations were performed?

Thanks for this important comment. In the present study, we investigated differences in gene expression profiles in skeletal muscle of endurance runners and control subjects using microarray analysis and quantitative RT-PCR (qRT-PCR). Thus, power calculations were hard to perform, because we planned to compare gene expression levels of many candidate genes. Thus, we referred to our previous paper (Kawaguchi et al. JCEM 2014), which compared candidate gene expression levels (same candidate genes shown in figure 4 in the present study) between 8 insulin sensitive and 9 insulin resistant subjects. In our previous study, we observed significant differences between the groups in several candidate gene expression levels. Given that muscle biopsy is risky for long-distance runners, we suppose that 7 endurance runners and 8 control subjects could be acceptable number of subjects.

There appear a lack of appreciation/mention that the athlete's paradox is not always evident ie. that IMCL is not always raised in athletes (eg. Bruce et al. 2003 JCEM; Bergman et al. 2018 Am J Physiol Endocrinol Metab) - and the results of this study are in agreement with this. An early study suggested IMCL was raised due to differences in fibre type (van Loon et al. 2004 Am J Physiol Endocrinol Metab), where type 1 fibres are known to have significantly higher IMCL. Also, given the differences between muscles, then the gene expression data should ideally be from the same muscle group.

We agree that we did not sufficiently explain the athlete’s paradox. Thus, we added all suggested references in the revised manuscript (Line 226-227). We also agree that gene expression level was evaluated in vastus lateralis muscle while IMCL was measured in TA and SOL. We added this as limitation (Line 230-231).

Intramyocellular lipid methodology - there appears excessive references which repeat or do not contain full methods. It is still unclear - was this done on a 1.5T scanner? Given that a long TE of 136ms and short TR of 1500ms were used, then these measures of IMCL and Cr would be T1 and T2-weighted. Given that exercise training likely results in muscle microstructural differences that could result in different relaxation times in athletes and controls, then were T1 and T2 measured and corrected for? Also, how was 'signal intensity' measured - details of fitting procedure should be documented.

Thanks for this comment. We added the protocol for MRS measurement in detail as below (Line87-95).

IMCL in the tibialis anterior muscle (TA) and soleus muscle (SOL) were measured after overnight fasting using 1H-magnetic resonance spectroscopy (VISART 1.5T EX V4.40; Toshiba, Tokyo), as described previously (1,16-21). Voxels (1.2 × 1.2 × 1.2 cm3) were positioned in the TA or SOL avoiding visible interfascial fat and blood vessels. Imaging parameters were set as follows; repetition time of 1500 msec, echo time of 136 msec, acquisition numbers of 192, and 1024 data points over a 1000-kHz spectral width. After examination, the resonances were line fit using a mixed Lorentzian/Gaussian function. After correction for T1 and T2 relaxations, the methylene signal intensity (S-fat), with peaks being observed at approximately 1.25 parts/million (ppm) was quantified. IMCL was quantified by S-fat and the creatine signal at 3.0 ppm (Cre) as the reference and was calculated as a ratio relative to Cre (S-fat/Cre).

Minor points:

Fig 4 legend - should 'HR' be athletes?

We revised the legend (Line200,202).

Fig 1 - would be better to have as dot plots to be able to view the distribution.

We revised the Fig 1 as suggested.

Line 59 - it is normal to refer to surnames, not forenames.

We corrected (Line59).

Reviewer 2 Report

This is a very solid paper, I just have one suggestion:
For figure 3, the authors can add a table to summarize gene expression changes in the microarray, the table should just focus on lipid metabolism. The added table should contain the gene symbol, p-value, and fold change. This modification can make figure 3 more informative and easy to understand.

Author Response

Responses to the comment of Reviewer 2

First of all, we would like to express our sincere thanks to you for identifying areas that needed corrections or modification. We would like to respond to your comment as described below.

This is a very solid paper, I just have one suggestion:
For figure 3, the authors can add a table to summarize gene expression changes in the microarray, the table should just focus on lipid metabolism. The added table should contain the gene symbol, p-value, and fold change. This modification can make figure 3 more informative and easy to understand.

Thanks for this comment. We added a table to summarize gene expression changes in the microarray as Supplementary Table 6.

Round 2

Reviewer 1 Report

I thank the authors for politely responding to my queries. However, unfortunately as the study aim was to compare results between control and athlete groups, the similar VO2max values between these groups suggests that the controls were also quite athletic (and/or athletes not extreme). This renders any subsequent results in primary outcomes (intramyocellular lipid ....) inconclusive, especially without power calculations.

Author Response

Responses to the comment of Reviewer 1

First of all, we would like to express our sincere thanks to you for identifying areas that needed corrections or modification. We would like to respond to your comment as described below.

I thank the authors for politely responding to my queries. However, unfortunately as the study aim was to compare results between control and athlete groups, the similar VO2max values between these groups suggests that the controls were also quite athletic (and/or athletes not extreme). This renders any subsequent results in primary outcomes (intramyocellular lipid ....) inconclusive, especially without power calculations.

Thanks for this reviewer’s comment. There was no significant difference in maximum oxygen uptake between the groups, thus, if we included more unfit subjects as control, we might have different results in IMCL levels and gene expression profile. However, muscle function is not merely associated with maximum oxygen uptake, but current training status; thus, we could observe some differences between the groups in the present study.

                 Accordingly, we mentioned about this limitation in the revised manuscript (Line 274-276).